# Identification of Mycoparasitism-Related Genes against the Phytopathogen *Botrytis cinerea* via Transcriptome Analysis of *Trichoderma harzianum* T4

**DOI:** 10.3390/jof9030324

**Published:** 2023-03-06

**Authors:** Yaping Wang, Xiaochong Zhu, Jian Wang, Chao Shen, Wei Wang

**Affiliations:** State Key Laboratory of Bioreactor Engineering, East China University of Science and Technology, Shanghai 200237, China

**Keywords:** *Trichoderma harzianum* T4, mycoparasitism, transcriptome, signal transduction proteins, carbohydrate active enzymes, hydrolytic enzymes, transporters, antioxidant enzymes

## Abstract

*Trichoderma harzianum* is a well-known biological control agent (BCA) that is effective against a variety of plant pathogens. In previous studies, we found that *T. harzianum* T4 could effectively control the gray mold in tomatoes caused by *Botrytis cinerea*. However, the research on its biocontrol mechanism is not comprehensive, particularly regarding the mechanism of mycoparasitism. In this study, in order to further investigate the mycoparasitism mechanism of *T. harzianum* T4, transcriptomic sequencing and real-time fluorescence quantitative PCR (RT-qPCR) were used to identify the differentially expressed genes (DEGs) of *T. harzianum* T4 at 12, 24, 48 and 72 h of growth in the cell wall of *B. cinerea* (BCCW) or a sucrose medium. A total of 2871 DEGs and 2148 novel genes were detected using transcriptome sequencing. Through GO and KEGG enrichment analysis, we identified genes associated with mycoparasitism at specific time periods, such as encoding kinases, signal transduction proteins, carbohydrate active enzymes, hydrolytic enzymes, transporters, antioxidant enzymes, secondary metabolite synthesis, resistance proteins, detoxification genes and genes associated with extended hyphal longevity. To validate the transcriptome data, RT-qCPR was performed on the transcriptome samples. The RT-qPCR results show that the expression trend of the genes was consistent with the RNA-Seq data. In order to validate the screened genes associated with mycoparasitism, we performed a dual-culture antagonism test on *T. harzianum* and *B. cinerea*. The results of the dual-culture RT-qPCR showed that 15 of the 24 genes were upregulated during and after contact between *T. harzianum* T4 and *B. cinerea* (the same as BCCW), which further confirmed that these genes were involved in the mycoparasitism of *T. harzianum* T4. In conclusion, the transcriptome data provided in this study will not only improve the annotation information of gene models in *T. harzianum* T4 genome, but also provide important transcriptome information regarding the process of mycoparasitism at specific time periods, which can help us to further understand the mechanism of mycoparasitism, thus providing a potential molecular target for *T. harzianum* T4 as a biological control agent.

## 1. Introduction

*Botrytis cinerea* (Ascomycota) is a necrotrophic plant pathogen with a wide range of hosts (>200 species) [1]. *Botrytis cinerea* can cause serious loss of crops during production and post-harvesting, the value of which can reach up to USD 1 billion per year [2,3], and for this reason, it is listed as the second most destructive pathogenic fungus in the world [4]. Today, chemical prevention is the main strategy used worldwide to control fungal diseases [5]. However, long-term use of chemical pesticides has led to human and animal health hazards, severe pesticide residues, environmental pollution and microbial resistance [6,7]. Therefore, it is essential to develop novel control methods that are environmentally friendly.

Biocontrol agents are an environmentally friendly alternative to chemical pesticides and have numerous advantages over the use of fungicides, such as cost-effectiveness, safety and environmental protection [8,9]. Strains of *Trichoderma* are excellent biological control agents and represent some of the most widely used fungi, playing a vital role in agriculture [10,11].

In a previous study, our research team isolated and purified from soil a biocontrol strain of *T. harzianum* T4 with a high application potential. This strain can effectively control the plant disease caused by *B. cinerea* [12], promote plant growth and increase crop yield [13,14]. In our previous study, we optimized the culture conditions for *T. harzianum* T4 and improved the yield of the *T. harzianum* biocontrol agent [15]. However, the research on its biocontrol mechanism is not comprehensive, particularly the mechanism of mycoparasitism. It has been shown that the mycoparasitism process mainly includes decomposition of the host cell wall by the secretion of hydrolytic enzymes; these processes are regulated by signal transduction proteins (heterotrimeric G protein and mitogen-activated protein kinase, MAPK) [16,17,18]. However, the mechanism of mycoparasitism is affected by the types of pathogen and *Trichoderma* isolates [19,20]. Therefore, it is necessary to study different strains to increase our understanding of the mycoparasitism of *T. harzianum*.

In previous studies, high-pressure induction of inactivated hyphae was used to simulate the presence of a fungal host in order to study the genes associated with fungal mycoparasitism [19,21,22,23]. In this study, using RNA-Seq, we identified the DEGs in the *T. harzianum* T4 strain after 12, 24, 48 and 72 h of culturing in a high-pressure-induced liquid medium containing inactivated *B. cinerea* cell wall (BCCW), and used the Czapek-Dox broth (CDB) medium inoculated only with *T. harzianum* as a control. This study helps to determine a large number of the DEGs involved in specific stage functions at different time periods and enhances understanding of the mycoparasitism of *T. harzianum* T4. We identified a large number of genes that are strongly associated with mycoparasitism. These genes are significantly upregulated in a specific time period and have the functions of coding for signal transduction (G protein and MAPK kinase), carbohydrate-active enzymes (especially β-glucosidase), transporters, antioxidant proteins, resistance to stress, detoxification, delaying hyphal senescence and toxic secondary metabolites. Finally, quantitative RT-PCR was used to verify the differential expression of candidate genes for *T. harzianum* T4 antagonism against *B. cinerea* via mycoparasitism in both transcriptome and dual culture.

## 2. Materials and Methods

### 2.1. Source of Fungal Strains

The wild-type *T. harzianum* T4 strain was stored in our laboratory (East China University of Science and Technology, Shanghai, China). *Botrytis cinerea* was obtained from the China General Microbial Species Preservation and Management Center (CGMCC, Beijing, China).

### 2.2. Culture Conditions

*Trichoderma harzianum* T4 was inoculated on a potato dextrose agar (PDA) plate and cultured for 3–5 days in a constant-temperature incubator at 28 °C. When the conidia had spread all over the plate, the conidia collected from the culture were washed repeatedly with 5 mL of sterile water and the spore suspension was diluted to 1 × 10^7^ for later use. Next, 1 mL of 1 × 10^7^ *T. harzianum* T4 conidia suspension was added to a fresh potato dextrose broth (PDB) liquid culture medium and the conidia were germinated in a rotating shaker at 28 °C for 12 h at 180 rpm. The grown mycelium was centrifuged at 4000 rpm for 10 min, collected and washed three times with sterile distilled water for subsequent induction experiments. The grown mycelium was transferred to the Czapek-Dox broth (CDB) induction medium containing 5 g sucrose, 3 g NaNO_3_, 1 g K_2_HPO_4_, 0.5 g MgSO_4_, 0.5 g KCl, 10 mg FeSO_4_ and supplemented with 1% inactivated cell wall of *B. cinerea* (previously autoclaved at 120 °C for 20 min) (BCCW). The CDB induction medium inoculated only with *T. harzianum*-grown mycelium without BCCW was used as the control. All cultures were incubated at 180 rpm at 28 °C for 12, 24, 48 and 72 h and the mycelium was collected. Samples for each incubation period were collected three times, including all treatments. The mycelium was immediately frozen using liquid nitrogen for 20–30 min and stored at −80 °C until the RNA was extracted. All collected RNA samples were used for the following transcriptome sequencing and real-time fluorescence quantitative PCR experiments (RT-qPCR).

### 2.3. Total RNA Extraction, cDNA Synthesis and Sequencing

Total RNA was extracted from 24 samples using the TransZol up Plus RNA kit (TransGen Biotech, Beijing, China) according to the manufacturer’s instructions. RNA was quantified and verified for integrity using the Nano assay in an Agilent BioAnalyzer 2100 (Agilent Technologies, Santa Clara, CA, USA). A total of 1 µg RNA per sample was used as input material for RNA sample preparation. Sequencing libraries were generated using the NEBNext UltraTM RNA Library Prep Kit for Illumina (NEB, Ipswich, MA, USA). Briefly, mRNA was purified from total RNA using poly-T oligo-attached magnetic beads. Fragmentation was carried out using divalent cations under elevated temperature in NEB Next First Strand Synthesis Reaction Buffer (5X). First-strand cDNA was synthesized using random hexamer primer and M-MuLV reverse transcriptase. Second-strand cDNA synthesis was subsequently performed using DNA polymerase I and RNase H. Remaining overhangs were converted into blunt ends via exonuclease/polymerase activities. The AMPure XP system (Beckman Coulter, Beverly, CA, USA) was used to purify fragments and amplify them by PCR to prepare cDNA libraries. All cDNA libraries were sequenced on the Illumina platform (Illumina, San Diego, CA, USA) to generate paired-end reads. SOAPnuke v2.1.0 software (BGI, Shenzhen, China) [24] was used to filter and splice the original readings to remove low-quality and unmeasurable bases (denoted by N), so as to obtain clean readings. The genome of *T. harzianum* T4 (separately cultured without *B. cinerea* cell wall) was annotated (NCBI: PRJNA715015) and used as a reference genome in this study.

### 2.4. Analysis of DEGs

DESeq2 [25] was used to analyze the significance of differential gene expression in *T. harzianum* T4 between the treated group and the control group at different growth stages (12, 24, 48 and 72 h). The threshold condition for DEGs was set as *p*-adjust (*q*-value) ≤ 0.05 and |Log_2_FC| ≥ 1. GO and KEGG databases were used for the enrichment and annotation analyses of the DEGs. The enrichment analysis method was hypergeometric distribution, with a *q*-value ≤ 0.05 taken as the threshold screening condition for significant enrichment of the GO term and KEGG pathway.

### 2.5. RT-qPCR

Twelve DEGs were randomly selected for RT-qPCR validation using the Bio-Rad CFX 96 quantitative fluorescence PCR instrument (Bio-Rad, Hercules, CA, USA). Primer5 v5.00 software (Premier Biosoft International) was used to design the verification primers (Table 1). RNA was reverse transcribed into cDNA using a TUREscript 1st Stand cDNA SYNTHESIS Kit. Each 20 µL PCR reaction consisted of 10 µL of 2 × SuperNovaPCRMix (Dye), 0.5 µL of forward primer, 0.5 µL of reverse primer, 1 µL of cDNA template and 8 µL of nuclease-free water. The operating cycle conditions were 95 °C for 4 min (1 cycle), 95 °C for 15 s, then 60 °C for 30 s (39 cycles), and for the melt curve analysis the temperature was increased from 65 °C to 95 °C in increments of 0.5 °C for 5 s (1 cycle). The gene expression level was calculated according to the threshold period (CT) 2^−ΔΔCT^ method and the 18s rRNA and α-tubulin were used as internal references to normalize the amount of total RNA present in each reaction.

### 2.6. Analysis of the Expression of Genes Related to Mycoparasitism in the Dual Culture

Dual culture is the most effective way to verify the expression of genes associated with mycoparasitism in a specific culture time period. RT-qPCR was used to analyze the expression of 24 genes possibly related to mycoparasitism of *T. harzianum* T4 (Table 1). Using a hole punch, a disc with a diameter of 5 mm was cut from the *T. harzianum* T4 and *B. cinerea* plates as a dual-culture inoculum. *Trichoderma harzianum T4* and *B. cinerea* were inoculated 8 cm apart on opposite sides of PDA plate medium covered with cellophane. As a control, dual-culture confrontation assays were conducted following the same procedure described above, except that *T. harzianum* was challenged with itself. The antagonistic plate was cultured in a constant-temperature incubator at 28 °C for 5 days and *T. harzianum* T4 mycelium samples (3 replicates of each sample) were harvested at 3 growth stages (before contact (BC), during contact (C) and after contact (AC)) of the dual culture and these were then used for RT-qPCR. The experiment was conducted with three repetitions for each sample and results were statistical analyzed by one-way analysis of variance (ANOVA) with Duncan tests conducted by SPSS statistics software (V. 21.0, SPSS Inc., Chicago, IL, USA). All results at *p* < 0.05 were considered statistically significant.

## 3. Results

### 3.1. Data Analysis of the Transcriptome

In this experiment, we used RNA-Seq to screen the differentially expressed genes of *T. harzianum* T4 at different growth stages (12, 24, 48 and 72 h) in the presence of BCCW. As shown by the Venn diagram in Figure 1A, a total of 2871 differentially expressed genes were detected using transcriptome sequencing. Of these, 26 DEGs were differentially expressed at 4 growth stages and 2262 DEGs were differentially expressed in only 1 specific stage. Of all the DEGs, the maximum number of DEGs was 1412 (843 upregulated, 569 downregulated) at 12 h. At 24 h and 48 h there were 273 (164 upregulated, 109 downregulated) and 269 (124 upregulated, 145 downregulated), respectively. The number of DEGs at 72 h was 794 (293 upregulated, 501 downregulated) (Figure 1B). Based on the reference genome of *T. harzianum* T4, 2148 novel genes were identified, which significantly enriched the available genomic annotation information for the T4 strain.

### 3.2. GO Enrichment Analysis of DEGs

GO enrichment analysis of DEGs showed that at the different growth stages (12, 24, 48 and 72 h) of *T. harzianum* in the presence of BCCW, GO terms mainly focused on biological processes (metabolism and single-organ processes); cell components (cell membrane, cell membrane components, cells and cell components); and molecular functions (catalytic activity, molecular binding and transporter activity) (Figure 2). In order to further analyze the genes related to mycoparasitism, GO enrichment analysis was performed on the upregulated DEGs (Figure 3). The results showed that in the biological process, carbohydrate metabolism was significantly enriched at 12, 24 and 48 h. The oxidation-reduction process and single organism metabolic process were significantly enriched at 72 h. In addition, kinase activity, phosphorylation, catalytic activity, hydrolase activity, transporter activity, cofactor binding, transmembrane transport activity, metal ion binding and oxidoreductase activity involved in the molecular process were all significantly enriched in different culture time periods of *T. harzianum* grown in the presence of BCCW.

### 3.3. KEGG Enrichment Analysis of DEGs

The KEGG enrichment analysis of the DEGs in *T. harzianum* grown for 12, 24, 48 and 72 h in the presence of BCCW showed that 25 KEGG pathways (*q* ≤ 0.05) were significantly enriched (Table 2). These pathways are mainly involved the metabolism of carbohydrates, lipids and amino acids, as well as the biodegradation and metabolism of exogenous substances and the metabolism of terpenoids and polyketides. To further analyze the genes associated with mycoparasitism, a KEGG enrichment analysis was performed on the upregulated DEGs. It showed the top 20 enrichment pathways of *T. harzianum* when grown for 12, 24, 48 and 72 h; of these pathways, 8 KEGG pathways were significantly enriched (*q* ≤ 0.05), involving carbohydrate metabolism and amino acid biosynthesis and degradation (Figure 4). Of these, starch and sucrose metabolism (ko00500) and amino sugar and nucleotide sugar metabolism (ko00520) were enriched at 12 h, 24 h and 48 h.

### 3.4. Analysis of Genes Related to T. harzianum Mycoparasitism Based on GO and KEGG Enrichment

#### 3.4.1. Kinase Activity and Signal Transducer Activity

In this experiment, we found that kinase activity-related and signal transducer activity-related genes were significantly upregulated at 12 h (Figure 5A). This mainly involves genes related to serine/threonine kinase, G protein signaling and MAPK kinase activity. We focused on 10 genes, including 7 kinase activity genes (log_2_FC > 2), of which 4 were kinase activity genes involved in glycolysis (scaffold 14.g95, scaffold 39.g6, scaffold 13.g198, scaffold 9.g267), and 3 were involved in protein serine/threonine kinase activities (scaffold 17. g109, scaffold 4.g70, novel.6707). The other three genes were related to the signal transduction activity: the guanine nucleotide-binding protein gamma subunit 1 gng-1 (scaffold 6.g186) and the guanine nucleus-binding protein alpha-2 subunit gna-2 (scaffold 6.g263) positively regulated the G protein, and the phosphate intermediate protein in YPD1 (scaffold 28.g81) regulated the MAPK signaling pathway.

#### 3.4.2. Carbohydrate Active Enzymes (CAZymes)

Based on KEGG and GO enrichment analysis, carbohydrate metabolism was significantly enriched at 12, 24 and 48 h. A large number of carbohydrate active enzymes, in particular, GO trem associated with hydrolase, were significantly enriched at 24 h and 48 h in response to the presence of BCCM. We statistically analyzed the DEGs of carbohydrate active enzymes (CAZymes) in *T. harzianum* T4 at 12, 24, 48 and 72 h (Figure 6A). There were 140 CAZyme genes with significant differential expression (Figure 6A). Large numbers of CAZyme-related genes were significantly upregulated at 12, 24 and 48 h, but not at 72 h. At 12 h, there were 62 differentially expressed genes, 42 of which were significantly upregulated. At 24 and 48 h, 24 and 26 CAZyme genes were upregulated, respectively. In this experiment, a total of 82 GH glycoside hydrolase differential genes, 23 coenzyme activity genes, 17 glycosyltransferase and carbohydrate esterase genes and 1 carbohydrate-binding module gene were significantly differentially expressed (Figure 6B).

Although the upregulated DEGs were significantly enriched in hydrolase-related genes at 24 and 48 h, the gene expression heat maps show that some hydrolase genes had an upregulated trend at 12 h. At 72 h, a large number of hydrolase-related genes were no longer upregulated or even significantly downregulated. This result suggests that hydrolase-related genes are unnecessary in the late growth phase of *T. harzianum* in the presence of BCCW. A large number of cellulase-related genes, such as beta glucosidase (scaffold 30.g33), endo-1,3 (4) beta glucosanase (scaffold 23.g102), exo-beta-1,3-glucanase (scaffold 11.g130), cellulase (scaffold 32.g55), and endochitinase 42 (scaffold 19.g145), were significantly upregulated at 12, 24 and 48 h. In addition, trehalose 6-phosphate synthase (scaffold 37.g2) genes were significantly upregulated at 12, 24 and 48 h and acetoxylan esterase (scaffold 32.g11) genes were significantly upregulated at 12 h and 48 h (Figure 5B).

#### 3.4.3. Transmembrane Transport

Transmembrane transport plays an essential role in fungal growth and development, as well as in mycoparasitism, and is responsible for the transport of carbon sources, nitrogen sources, metal ions and toxic substances. GO enrichment analysis showed that in *T. harzianum* in the presence of BCCW, the upregulated gene was significantly enriched in the GO term of the transmembrane transport activity at 48 h. Of the 22 significantly upregulated transporters (Figure 5C), 14 belonged to the major facilitator superfamily (MFS) of transporters, including 7 sugar transporter family genes (scaffold 37.g3, scaffold 22.g139, scaffold 17.g32, scaffold 16.g31, scaffold 12.g146, scaffold 10.g185 and scaffold 10. g155), 2 proton-dependent oligopeptide transporter (POT/PTR) genes (scaffold 81.g2 and scaffold 24.g29), 2 monocarboxylate transporter family genes (scaffold 9.g58 and scaffold 29.g70), and 1 iron transporter (scaffold 7.g253). In summary, MFS transporters play an influential role in *Trichoderma* mycoparasitism. Three amino acid polyamine organic (APC) family genes (scaffold 4.g305, scaffold 6.g129 and scaffold 11.g154) are responsible for the transport of amino acids and choline. The ammonium transporter MEP1 (scaffold 27.g12) is closely related to the nitrogen source transport and absorption of *Trichoderma* in the process of mycoparasitism. In general, transporter-related genes are involved in a series of material transports, such as carbon source, nitrogen source, amino acid, drug efflux protein and metal ion transport, during the growth and development of *T. harzianum* T4 in the presence of BCCW.

#### 3.4.4. Antioxidant Enzymes

Transcriptome sequencing of the *T. harzianum* T4 strain grown for 12, 24, 48 and 72 h in the presence of BCCM showed that the genes encoding oxidoreductase activity were significantly enriched in several culture time periods. Based on differential expression multiples, 22 differentially expressed genes that were significantly upregulated were identified (Figure 5D). The differentially expressed genes include genes involved in cellular detoxification, oxidase, peroxidase, osmotic pressure regulation and amino acid biosynthesis and metabolism. Of these, NADPH oxidase *Nox* (scaffold 2.g181, scaffold 3.g249), which is responsible for ROS production, was significantly upregulated at 12 h. *PRX1* (scaffold 13.g225), a bifunctional enzyme that can protect the body from oxidative damage, was significantly upregulated at 12 h. *AzaB* (scaffold 3. g68), which belongs to the polyketide synthase (PKS) family, was significantly upregulated at 24 h. *Glt1* (scaffold14 14.g81), a glutamate synthase precursor-related gene, was significantly upregulated at several culture times, which attracted our attention and is further elaborated on in the discussion below.

### 3.5. The Top 10 Significantly Upregulated Genes

To further evaluate the expression profiles of differentially expressed genes at different growth stages, the top 10 differentially expressed genes at 12, 24, 48 and 72 h were identified (Table 3). The top 10 significantly upregulated genes at 12 h were 3 antistress and detoxification-related oxidoreductases (*noxA*, *SPAC513.07* and *AIFM2*), 3 amino acid biosynthesis or metabolism enzymes (*pha*, *2QPCT* and *ALDH6A1*), 2 glycoside hydrolase genes (*bgn13.1* and *abf1*), and 1 peptidase gene (*FUB8*). The top 10 significantly upregulated genes at 24 h were 6 glycoside hydrolases (*neg-1*, *bgn13.1*, *ARB_02077*, *cel3A*, *eng2* and *chi2*), 1 carbohydrate esterase (*CUTA*), 1 protease (*prb1*), 1 amino acid biosynthesis enzyme (*mfnA*), 1 WSC domain-containing protein and 1 uncharacterized protein (*YEL023C*). The top 10 significantly upregulated genes at 48 h were 3 glycoside hydrolases (*bgn13.1*, *neg-1* and *cel3A*), 1 carbohydrate esterase (*CUTA*), 1 protease (*prb1*), 1 oxidoreductase (*arsH*), 1 aminotransferase (*gatA*), 1 transmembrane transport protein (*apdF*), 1 WSC domain-containing protein and 1 uncharacterized protein (YEL023C). The top 10 significantly upregulated genes at 72 h were 2 hydrolases (*bgn13.1*, *SPAC5H10.01*), 2 transmembrane transporters (*FRE7*, *apdF*), 1 ribosomal protein (*rpl-3*), 1 WSC domain-containing protein, 1 aromatic compound metabolism (*hpcH*), 1 amino transferase (*gatA*), 1 osmoregulation-related gene (*gpd1*) and 1 HMG-CoA lyase-family gene (*liuE*). Under simulated fungal mycoparasitism conditions, antistress- and detoxification-related genes were preferentially expressed at 12 h and a combination of glycoside hydrolases, cutinases and proteases were preferentially expressed at 24 and 48 h, with the highest number of glycoside hydrolases appearing at 24 h (6 out of 10). A new gene encoding endo-1,3 (4) beta glucanase (novel.5121) was significantly upregulated at 12, 24, 48 and 72 h. A gene encoding the WSC domain-containing protein (scaffold 23.g56) was significantly upregulated at 24, 48 and 72 h, and its main functions require further research and exploration.

### 3.6. Real-Time Quantitative PCR Verification (RT-qPCR)

Twelve upregulated differentially expressed genes were randomly selected from the 12, 24, 48 and 72 h transcriptome data for RT qPCR validation (Figure 3). These were gluco endo-1,3-beta-glucosidase (*bgn13.1*), gluco 1,3-beta-glucosidase (*EXG1*), cyclochrome P450 55A1 (*CYP55A2*), beta glucosidase (*bglA*), glutamate/leucine/phenylalanine/valine dehydrogenase (*GDH2*), hexokinase (*glkA*), D-lactate dehydrogenase (DLD1), 4-aminobutyrate aminotransfer (*gatA*), pyruvate decarboxylase (pdcA), glyceraldehyde-3-phosphate dehydrogenase (*gpd1*), aspartyl-tRNA synthetase (*DPS1*) and sedoheptulose-1,7-bisphophase (*E3.1.3.37*). The RT-qPCR results show that the expression trends of the genes were basically consistent with the RNA-Seq data (Figure 7).

### 3.7. Dual-Culture of T. harzianum T4 and B. cinerea to Verify the Genes Related to Mycoparasitism

In order to further verify the genes in *T. harzianum* related to mycoparasitism, we analyzed the expression patterns of these genes using a double-culture direct confrontation experiment [26,27]. RT-qPCR was performed using the total RNA of samples of the *T. harzianum* and *B. cinerea* dual culture before (BC), during (C) and after (AC) physical contact. We selected 24 genes that may be associated with mycoparasitism for analysis: 5 kinases and signal transduction-related genes (*SAT4*, *glkA*, *PFK26*, *mpr1* and *gng-1*), 7 glycoside hydrolytic enzymes (*Pc12g07500*, *eglC*, *blr3397*, *btgC*, *EXG1*, *chit46* and *bglA*), 1 acid phosphatase (*aphA*), 2 transporters (*VPS33* and *apdF*), 1 peroxidase (*PRX1*), 2 dehydrogenases (*mtlD* and *GDH2*), 1 decarboxylase (*pdcA*), 1 transferase (*GST*), 2 amino acid biosynthesis genes (*LEUA* and *glt1*) and trehalose and tetrapyrrole biosynthesis genes (*TPS2* and *HEM2*) (Figure 8). *Trichoderma harzianum* was grown with itself as a control. Based on the results of dual-culture RT-qPCR, 15 of the 24 genes were upregulated during and after contact between *T. harzianum* T4 and *B. cinerea*, providing further evidence that these genes are important factors in mycoparasitism.

## 4. Discussion

Some studies have confirmed that liquid media containing fungal host cell walls can be used as a model to identify the genes related to mycoparasitism [21,22,23,28,29,30]. The mechanism of mycoparasitism is a complex process, which is regulated by multiple factors such as different isolated strains and pathogens. In this study, we used transcriptome methods to explore the genes related to the mycoparasitism of *T. harzianum* T4 at the 4 growth stages of 12, 24, 48 and 72 h in the presence of BBCW or sucrose. A total of 10,266 predicted genes were identified from our transcriptome data, including 2871 DEGs and 2148 novel genes, greatly enriching the available genome annotation information for *T. harzianum* T4. Through GO and KEGG enrichment analysis of DEGs, we found that carbohydrate metabolism was significantly enriched at 12, 24 and 48 h. At 24 and 48 h, the *T. harzianum* T4 strain upregulated most of the genes related to carbohydrate active enzymes (in particular, hydrolase genes) in response to the presence of BCCW. By enriching the upregulated DEGs at different culture times, we further analyzed the genes that were significantly enriched at specific epochs, such as kinases and signal transduction genes, CAZymes (in particular, hydrolase), transmembrane transporters and oxidoreductase. This discovery confirmed that *T. harzianum* could maintain its own growth and eliminate the host through a series of processes, such as upregulating the expression of carbohydrate activity enzyme-related genes, enhancing the activity of hydrolase to hydrolyze the host cell wall and regulating the method of nutrition acquisition.

The mycoparasitism process of *T. harzianum* involves a series of activities such as recognition, adherence, attachment, persistence, hydrolysis and parasitism of the host hyphae at different growth stages [31]. In this regard, MAPK and G protein have been proven to be involved in the recognition and attachment stages of mycoparasitism [18,32]. Recent studies have shown that MAPK participates in the formation of cellulase and regulates its expression at the transcription level [33,34,35]. From the transcriptome of *T. harzianum* T4, we found that two genes encoding G protein and three MAPK protein kinases were significantly upregulated under BCCM induction compared with the control. In addition, an opportunistic 2-component system, phosphorelay intermediate protein YPD1 (mpr1), was significantly upregulated in the presence of BCCM at 12 h, which could positively regulate the MAPK cascade [36]. YPD1 also plays a key role in cell functions such as cell wall synthesis and spore formation, osmotic and oxidative stress, and toxicity [37,38].

CAZymes are closely related to the nutritional patterns of fungi and participate in the degradation of the cell walls of pathogens and stored compounds [39,40]. When the *T. harzianum* T4 strain was grown in an induction medium with BCCW for 24 and 48 h, a significant number of hydrolase genes, such as chitinase, cellulase and glucanase, were significantly upregulated. In addition to hydrolase genes, glycosyltransferase and protease genes were significantly upregulated, and these play a key role in mycoparasitism [41,42]. Cellulase can hydrolyze the mycelia of fungal pathogens during the mycoparasitism process. Some studies have shown that the greater the cellulase activity, the greater the biological control potential [43,44,45]. A new gene encoding endo-1,3 (4) beta glucanase (*bgn13.1*) was significantly upregulated in multiple culture time periods and this was directly involved in the mycoparasitism between *T. harzianum* and *B. cinerea* [46,47]. The role of these upregulated genes in the mycoparasitism of *T. harzianum* T4 involves the penetration and degradation of the cell wall, which inhibits the spore germination and mycelial growth of the pathogen.

*Trichoderma harzianum* T4 adapted to the presence of the host cell walls by upregulating genes associated with CAZymes in carbohydrate metabolism, particularly glycosidic hydrolases. After hydrolyzing carbohydrate macromolecules, transport and absorption also play a role in mycoparasitism. The upregulation of the glucose transporter HXT at 12 h was closely related to improvements in the sexual reproduction, antioxidant capacity, pathogenicity and toxicity of *T. harzianum* T4 [48,49,50]. The POT/PTR family protein PTR2 was upregulated at 24 and 48 h, is involved in the transport of small peptides and is closely related to the internal transport of nutrient stores and the absorption of host surface decomposition products during mycoparasitism [51]. MEP1, an ammonium transporter that can absorb low concentrations of ammonium under nitrogen restriction to maintain cell growth, was upregulated at 12, 24, 48 and 72 h [52]. SIT1, which mediates iron absorption, was upregulated at 48 h and is essential to *T. harzianum* mycoparasitism [53,54,55]. The upregulation of choline transporter HNM1 at 48 h and 72 h is critical for conidial germination, sprout tube elongation and fungal toxicity [56]. Therefore, transmembrane transporters participate in the transport of a range of materials, such as carbon and nitrogen sources, metal ions and toxic substances, that play an important role in the process of mycoparasitism.

We found that *T. harzianum* responded to the oxidative damage by upregulating antioxidant enzymes or metabolites during stress conditions (in the presence of the host or at the late phase of cultivation) [57]. Transcriptome data revealed that *T. harzianum* responded to BCCW by upregulating NADPH oxidase NOxA to produce ROS while upregulating peroxidase Prx1 to protect cells from oxidative damage [58,59]. In addition, *MtlD* and *P5CR* were significantly upregulated; these genes participate in the biosynthesis of mannose and proline and play a key role in regulating cellular osmolarity and oxidative stress [60,61]. Of these, mannitol is a powerful quencher of reactive oxygen species (ROS) in fungi [62]. *T. harzianum* directly attacks the host during mycoparasitism and produces toxic secondary metabolites to kill the pathogen. Of these, nonribosomal peptide synthetases (NRPSs) and lovastatin nonaketotide syntheses AzaB (a polyketone compound) were significantly upregulated. They are important secondary metabolites of *T. harzianum* and these genes can be combined with hydrolase genes to further inhibit the growth of pathogens and enhance the mycoparasitism of *T. harzianum* T4 [63,64].

When faced with host and secondary metabolites, *T. harzianum* upregulated resistance proteins and detoxification-related genes to increase its ability to parasitize by enhancing its adaptation to stress environments. RP-L3e is a *Trichoderma*-resistant protein gene which plays an important role in amino acid tRNA binding, peptidyltransferase activity and drug resistance [65]. GST is a unique multifunctional cell membrane enzyme that is involved in the detoxification of toxic compounds and protection from oxidative damage [66,67,68,69,70]. An aging-related NAD-dependent histone deacetylase, SIR2, regulated by S-glutathione, extends lifespan by increasing lipid droplet and trehalose content and plays a key role in fungal parasites [71,72]. We also found that the expression of lipid droplet-associated hydrolase was downregulated at 12 h. TPS involved in trehalose synthesis was upregulated at 12, 24 and 48 h. Trehalose formation is essential for the maintenance of cell homeostasis and formation of appressoria [73,74].

In addition, a gene encoding a WSC domain-containing protein attracted our attention. The gene was significantly upregulated at 24, 48 and 72 h of *T. harzianum* growth in the presence of BCCM. Some studies have shown that the WSC domain-containing protein can activate the MAPK cascade [75], which is related to the integrity of cell walls, oxidation, hypertonic and metal ion binding. Studies have also shown that this protein is related to lectin, although this has not been definitively proved [76]. Two new genes encoding transketolase, *TKL1* and *TKL2*, were significantly upregulated during mycoparasitism. *TKL1* is mainly responsible for the response to osmotic stress and promotion of biotrophic growth and its overexpression can increase lipid content. *TKL2* is mainly responsible for the response to oxidative stress [77]. These two transketolase enzymes play unique and complementary roles in coping with different environmental pressures [78]. However, the specific functions of these novel genes in the mycoparasitism of *T. harzianum* require verification by further experiments.

Among the DEGs in *T. harzianum* T4, we identified many unknown functional or hypothetical protein genes. These DEGs are highly induced by BCCW but are not, or are rarely, expressed when *T. harzianum* T4 is grown with itself, which may be significant in mycoparasitism. In our forthcoming work, we will further verify the specific functions of mycoparasitism-related genes screened by the transcriptome via gene knockout, overexpression and double-culture antagonism experiments.

## 5. Conclusions

In conclusion, we used the transcriptome sequencing technology to identify DEGs related to mycoparasitism of *T. harzianum* T4 at 12, 24, 48 and 72 h of growth in the presence of cell walls of *B. cinerea* (BCCW). Through GO and KEGG enrichment analysis of DEGs, we identified genes associated with mycoparasitism at specific time periods, such as genes encoding kinases, signal transduction proteins, carbohydrate active enzymes, hydrolytic enzymes, transporters, antioxidant enzymes, secondary metabolite synthesis, resistance proteins, detoxification and genes associated with the extension of hyphal longevity. The transcriptome data provided in this study will not only improve the annotation information of gene models in the *T. harzianum* T4 genome, but also provide important transcriptome information involved in the process of mycoparasitism at specific time periods, thus providing a potential molecular target for other genes that use *T. harzianum* T4 as a biological control agent.

## Figures and Tables

**Figure 1 jof-09-00324-f001:**
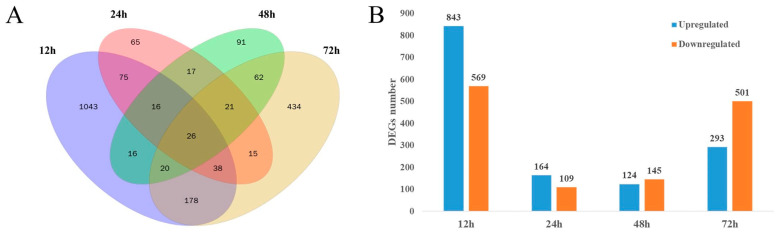
Statistical analysis of DEGs in *T. harzianum* T4 at different growth stages (12, 24, 48 and 72 h) in the presence of BCCW. (**A**) Venn diagram of DEGs. (**B**) Numbers of upregulated and downregulated DEGs.

**Figure 2 jof-09-00324-f002:**
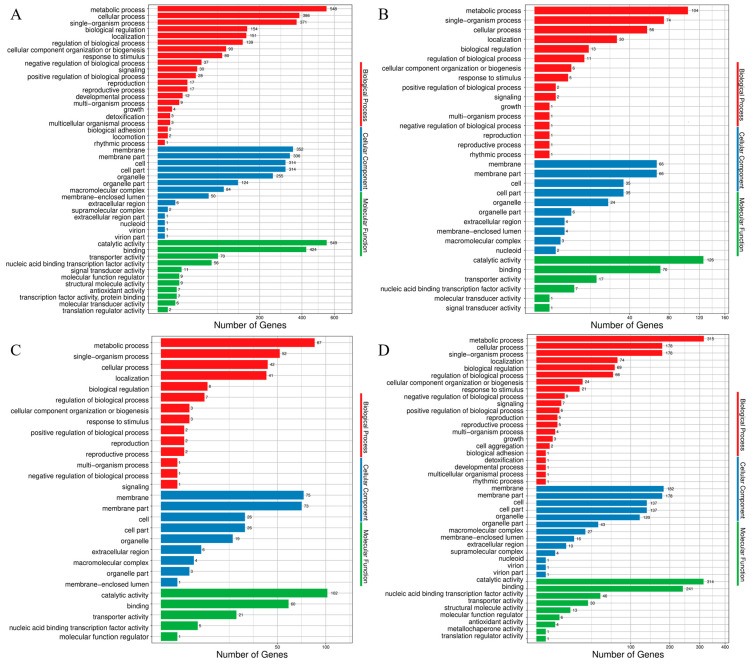
GO functional categories of all DEGs in *T. harzianum* T4 grown for (**A**) 12 h, (**B**) 24 h, (**C**) 48 h and (**D**) 72 h in the presence of BCCW.

**Figure 3 jof-09-00324-f003:**
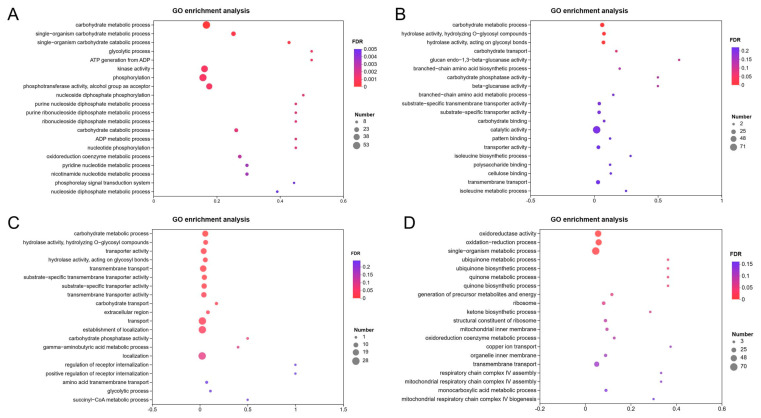
GO enrichment analysis bubble diagram showing all upregulated DEGs in *T. harzianum* T4 grown for 12 h (**A**), 24 h (**B**), 48 h (**C**), and 72 h (**D**) in the presence of BCCW. The ordinate represents the GO classification description, and the abscissa rich factor represents the ratio of differentially expressed genes of this term to all genes that were annotated in the term.

**Figure 4 jof-09-00324-f004:**
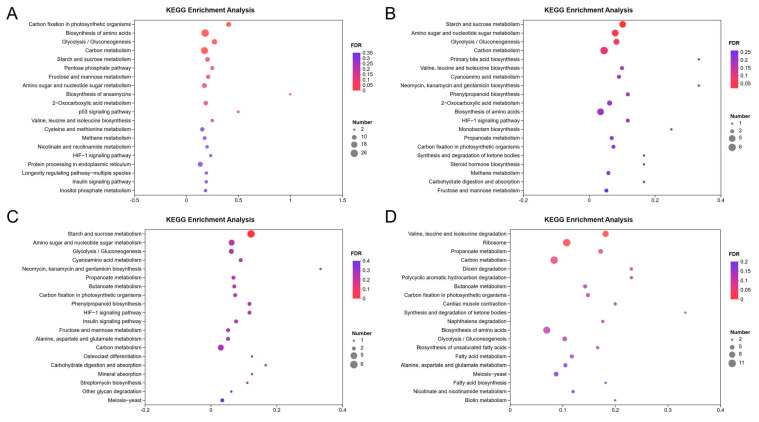
Bubble diagram of KEGG pathways of upregulated DEGs in *T. harzianum* T4 grown for (**A**) 12 h, (**B**) 24 h, (**C**) 48 h and (**D**) 72 h in the presence of BCCW.

**Figure 5 jof-09-00324-f005:**
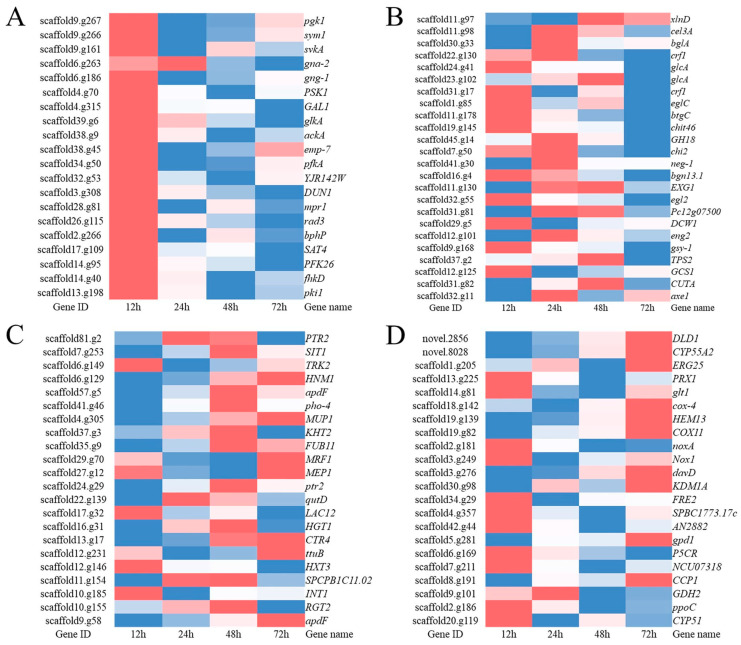
Heatmap of the DEGs involved in (**A**) signal transduction and kinase activity; (**B**) carbohydrate activity enzymes; (**C**) transmembrane transport and transporter; and (**D**) antioxidant proteins in *T. harzianum* T4 samples treated with BCCW at different time points (12, 24, 48 and 72 h).

**Figure 6 jof-09-00324-f006:**
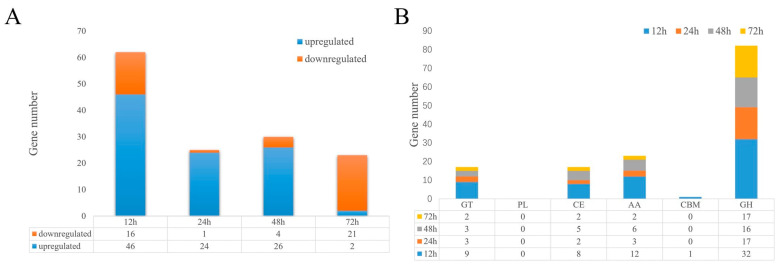
(**A**) Summary of DEGs of CAZymes in *T. harzianum* T4 grown for 12, 24, 48 and 72 h in the presence of BCCW. (**B**) Summary of upregulated and downregulated DEGs of CAZymes. GH, AA, CBM, CE, GT and PL represent glycoside hydrolase, auxiliary activity, carbohydrate-binding module, carbohydrate esterase, glycosyltransferase and polysaccharide lyase, respectively.

**Figure 7 jof-09-00324-f007:**
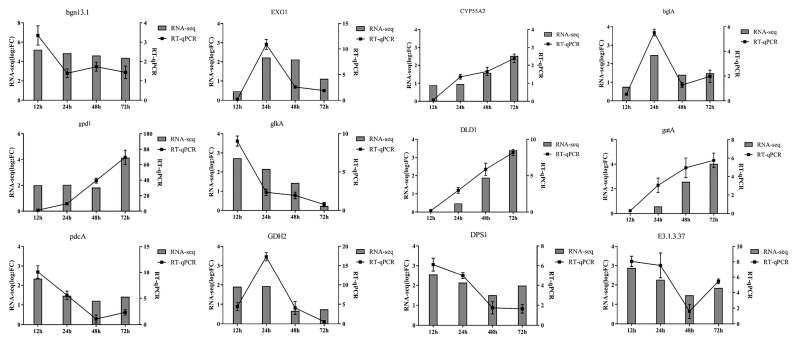
Comparison of RNA-Seq and real-time RT-PCR analysis. The relative expression levels of 12 single genes of *T. harzianum* T4 when treated with BCCW for different times (12, 24, 48 and 72 h) were compared with RNA-Seq in pairs. The histograms represent the relative expression level (log_2_FC) assessed by RNA-Seq. The black dotted lines indicate the relative expression level (2^−ΔΔCt^). The error bars represent the standard deviation of three duplicates.

**Figure 8 jof-09-00324-f008:**
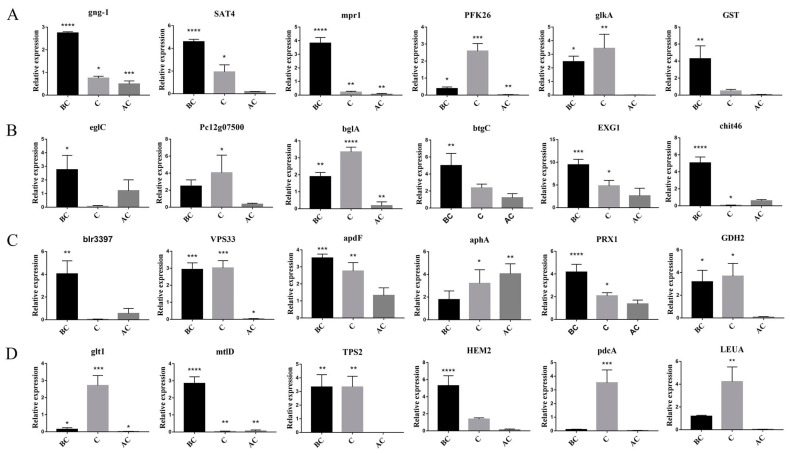
Differential expression and quantitative analysis of 24 genes related to mycoparasitism from the dual-culture direct confrontation experiment. BC: before contact, C: contact, AC: after contact. (**A**) Expression analysis of guanine nucleotide-binding protein gamma subunit 1, HAL protein kinase, phosphorelay intermediate protein, 6-phosphofructo-2-kinase, hexokinase and glutathione S-transferase. (**B**) Expression analysis of glucan endo-1,3-beta-glucosidase, murein transglycosylase, beta-glucosidase, glucan endo-1,3-beta-glucosidase, exo-beta-1,3-glucanase and endochitinase. (**C**) Expression analysis of nitrilase, lysine-specific histone demethylase 1, peptide chain release factor 1, acid phosphatase, peroxidase and lutamate/leucine/phenylalanine/valine dehydrogenase. (**D**) Expression analysis of glutamate synthase precursor, mannitol-1-phosphate dehydrogenase, trehalose 6-phosphate synthase, delta-aminolevulinic acid dehydratase, pyruvate decarboxylase and leucine-2. * *p* < 0.05, ** *p* < 0.01, *** *p* < 0.001, **** *p* < 0.0001.

**Table 1 jof-09-00324-t001:** Primer design for RT-qPCR validation experiments in *T. harzianum* T4.

Putative Function	Gene Name	Primers for qPCR (5′ to 3′)	
		Forward	Reverse
Glucan endo-1,3-beta-glucosidase	*bgn13.1*	CAGCAGTAGCAATGAATGTAAG	GAAGTCCTCAGCGATGTG
Cytochrome P450 55A1	*CYP55A2*	CATTGCCAGTTCCATCAT	CTTCCGTTGGTTCGTATG
D-lactate dehydrogenase	*DLD1*	GGACAGCATCAACCAGAC	CCGACCAGAGCGTATGAG
4-aminobutyrate aminotransferase	*gatA*	TGACTTCTCCTCCTGACAT	TGTTGAACTGGCGGTAAG
Pyruvate decarboxylase	*pdcA*	ACAACGACATCTCTAACTG	ATTCCTCCTTGGTCTTGA
Glyceraldehyde-3-phosphate dehydrogenase	*gpd1*	GTCTTGGTTGTCGCTGAG	CCCTTGTCCTTCCTCTGG
Aspartyl-tRNA synthetase	*DPS1*	TGCTGTTTCTTCGGATGA	CTTTCCAGATTCCCACAATAG
Sedoheptulose-1,7-bisphosphatase	*E3.1.3.37*	AGACACATGACCGAGTTC	AATAGCACGCCTTCCAATA
Mutanase precursor	*Pc12g07500*	CCGCCAATGTTGCTATTCGG	ATCATAGTCTGCCGAGCTGC
Glucan endo-1,3-beta-glucosidase	*eglC*	CGGCACTCTGGTCAACTACA	AAGGGTATGTGTCCATGCCG
Peroxidase	*PRX1*	CGACAACTGGGTCGTCTTCT	ATCCAGCCATTGTGGGAGTC
Acid phosphatase	*aphA*	GCGAATCCTCCGTTCTGGTT	GCAAAGTCCGTTTCCGTGTC
HAL protein kinase	*SAT4*	GCTGCAGATGCTCAACCCTA	TTGCCGCCGTAATCCTTCTT
Guanine nucleotide-binding protein gamma subunit 1	*gng-1*	ACTACGCGAGGATTTGGACC	CTTTGGCACGGTTCCCCATA
Nitrilase	*blr3397*	AAGTGAGGCATGGACGAAGG	TTCGTGATTGCTCTTCCCCC
Glutathione S-transferase	*GST, gst*	GCATCAGTAAGTTTGGCGGC	CGGTCAATGTACTCTCCCGC
Murein transglycosylase	*btgC*	GAGAGCGAACCTCTCGATGG	GAGGTTGATGCTGGTCGTGA
Exo-beta-1,3-glucanase	*EXG1*	CGCGGTCTGTTGGTTGAAAG	CTGGAATTGGGTTGCGTTGG
Glutamate/leucine/phenylalanine/valine dehydrogenase	*GDH2*	GGGATACCGTGTGCAGTTCA	TCTCGTTGTCAGACTTGCCC
Endochitinase 42	*chit46*	CAGACGGCACAGTTGTCTCT	CAGAGGGGAAGTTGGTGGAC
Beta-glucosidase	*bglA*	GACACTGCGATCCAGAACCA	GTAACCCTCACCACTGTCCG
Hexokinase	*glkA*	CGCTAGATCGAGACAGCGTT	GACCAGTCATTGGTGCTGGA
Aspyridones efflux protein	*apdF*	CACAACGAGTGTCTCGACCA	GACATGTTGGCAATCGTCGG
Leucine-2	*LEUA*	AAGCTGTCCGAGTACACTGC	GCATGCCGGTTGAATCACTC
Lysine-specific histone demethylase 1	*VPS33*	TCGGAGAAAGCGGATGCAAT	AAGAAGCGTCCCCGATTTGT
Osomolarity two-component system, phosphorelay intermediate protein YPD1	*mpr1*	TAAAGGTCCGAGACGGTTGC	AGACTTGACTGCTGTGAGCG
Trehalose 6-phosphate synthase	*TPS2*	ATGGCATGAACACGACCAGT	GGATTGCATCGCGGAGACTA
Delta-aminolevulinic acid dehydratase	*HEM2*	TGTCTGTGCCGCAGTACTAAT	GTCAGAGTAGATGCCGGGAG
Glutamate synthase precursor	*glt1*	AATGGGCTGCTTGCCAAATG	GAGGCGCAAATGATTCGGTC
6-phosphofructo-2-kinase	*PFK26*	ATTGAGCGCATCACTGACCA	AATGCCATAAGGCTTGGGCT
Pyruvate decarboxylase	*pdcA*	CAAGTACCTCCGAGCTGCAA	CTCTTCGTTGACAGCACCCT
Mannitol-1-phosphate dehydrogenase	*mtlD*	GCAACCCTCACCTGGAAGAC	CTGGAAGCGGAACGTCATCT
18s rRNA	*18s rRNA*	CAACCATAAACGATGCCGA	AGCCTTGCGACCATACTCC
*α*-tubulin	*α-tubulin*	TATCTGCTACCAGGCTCCCGAGAA	TGGTGTTGGACAGCATGCAGACAG

**Table 2 jof-09-00324-t002:** Significantly enriched KEGG pathways (*q* ≤ 0.05) of DEGs in *T. harzianum* T4 grown for 12, 24, 48 and 72 h in the presence of BCCW.

Culture Time	Regulated	Pathway Hierarchy1	Pathway Hierarchy2	KEGG Pathway	Pathway ID	Gene Number	Background Number	*q*-Value
12 h	Up	Metabolism	Energy metabolism	Carbon fixation in photosynthetic organisms	ko00710	11	27	1.46 × 10^−4^
		Metabolism	Overview	Biosynthesis of amino acids	ko01230	26	143	4.15 × 10^−4^
		Metabolism	Carbohydrate metabolism	Glycolysis/Gluconeogenesis	ko00010	13	48	9.70 × 10^−4^
		Metabolism	Overview	Carbon metabolism	ko01200	23	131	1.36 × 10^−3^
	Down	Cellular processes	Transport and catabolism	Peroxisome	ko04146	20	53	1.66 × 10^−11^
		Metabolism	Amino acid metabolism	Valine, leucine and isoleucine degradation	ko00280	14	44	6.14 × 10^−7^
		Metabolism	Lipid metabolism	Fatty acid degradation	ko00071	8	28	1.65 × 10^−3^
		Metabolism	Lipid metabolism	Primary bile acid biosynthesis	ko00120	3	3	2.86 × 10^−3^
		Metabolism	Xenobiotics biodegradation and metabolism	Benzoate degradation	ko00362	5	11	2.86 × 10^−3^
		Metabolism	Carbohydrate metabolism	Propanoate metabolism	ko00640	7	29	8.06 × 10^−3^
		Metabolism	Amino acid metabolism	Phenylalanine metabolism	ko00360	7	38	3.82 × 10^−2^
24 h	Up	Metabolism	Carbohydrate metabolism	Starch and sucrose metabolism	ko00500	5	49	2.12 × 10^−2^
		Metabolism	Carbohydrate metabolism	Amino sugar and nucleotide sugar metabolism	ko00520	5	63	3.41 × 10^−2^
	Down	Metabolism	Amino acid metabolism	Valine, leucine and isoleucine degradation	ko00280	11	44	8.28 × 10^−11^
		Metabolism	Carbohydrate metabolism	Propanoate metabolism	ko00640	5	29	6.10 × 10^−4^
		Metabolism	Metabolism of other amino acids	Taurine and hypotaurine metabolism	ko00430	3	10	3.77 × 10^−3^
		Metabolism	Lipid metabolism	Fatty acid degradation	ko00071	4	28	3.78 × 10^−3^
		Metabolism	Xenobiotics biodegradation and metabolism	Benzoate degradation	ko00362	3	11	3.78 × 10^−3^
		Metabolism	Amino acid metabolism	Tryptophan metabolism	ko00380	5	59	5.59 × 10^−3^
		Metabolism	Amino acid metabolism	Tyrosine metabolism	ko00350	4	33	5.59 × 10^−3^
		Metabolism	Xenobiotics biodegradation and metabolism	Aminobenzoate degradation	ko00627	4	38	7.79 × 10^−3^
		Metabolism	Lipid metabolism	Synthesis and degradation of ketone bodies	ko00072	2	6	1.38 × 10^−2^
		Metabolism	Amino acid metabolism	Arginine biosynthesis	ko00220	3	25	1.90 × 10^−2^
		Metabolism	Xenobiotics biodegradation and metabolism	Styrene degradation	ko00643	3	27	2.16 × 10^−2^
48 h	Up	Metabolism	Carbohydrate metabolism	Starch and sucrose metabolism	ko00500	6	49	1.75 × 10^−3^
	Down	Metabolism	Biosynthesis of other secondary metabolites	Isoquinoline alkaloid biosynthesis	ko00950	3	11	2.63 × 10^−3^
		Metabolism	Amino acid metabolism	Phenylalanine metabolism	ko00360	4	38	3.32 × 10^−3^
		Metabolism	Amino acid metabolism	Glycine, serine and threonine metabolism	ko00260	4	46	4.71 × 10^−3^
		Metabolism	Amino acid metabolism	Tryptophan metabolism	ko00380	4	59	9.24 × 10^−3^
		Metabolism	Amino acid metabolism	Tyrosine metabolism	ko00350	3	33	1.31 × 10^−2^
		Metabolism	Biosynthesis of other secondary metabolites	Tropane, piperidine and pyridine alkaloid biosynthesis	ko00960	2	8	1.31 × 10^−2^
72 h	Up	Metabolism	Amino acid metabolism	Valine, leucine and isoleucine degradation	ko00280	8	44	5.56 × 10^−3^
		Genetic information processing	Translation	Ribosome	ko03010	11	102	1.63 × 10^−2^
	Down	Metabolism	Lipid metabolism	Fatty acid degradation	ko00071	8	28	2.43 × 10^−5^
		Cellular processes	Transport and catabolism	Peroxisome	ko04146	9	53	2.44 × 10^−4^
		Metabolism	Amino acid metabolism	Tryptophan metabolism	ko00380	8	59	3.03 × 10^−3^
		Metabolism	Overview	Fatty acid metabolism	ko01212	6	34	4.03 × 10^−3^
		Metabolism	Metabolism of terpenoids and polyketides	Geraniol degradation	ko00281	2	4	4.42 × 10^−2^
		Metabolism	Xenobiotics biodegradation and metabolism	Dioxin degradation	ko00621	3	13	4.42 × 10^−2^
		Metabolism	Xenobiotics biodegradation and metabolism	Polycyclic aromatic hydrocarbon degradation	ko00624	3	13	4.42 × 10^−2^
		Organismal Systems	Endocrine system	Adipocytokine signaling pathway	ko04920	3	13	4.42 × 10^−2^
		Metabolism	Amino acid metabolism	Valine, leucine and isoleucine degradation	ko00280	5	44	4.52 × 10^−2^
		Metabolism	Carbohydrate metabolism	Amino sugar and nucleotide sugar metabolism	ko00520	6	63	4.52 × 10^−2^

**Table 3 jof-09-00324-t003:** Log_2_ fold change of the top 10 DEGs in *T. harzianum* T4 treated with BCCW at 12, 24, 48 and 72 h.

Times of Induction	Gene ID	Gene Name	Putative Function	Log_2_FC
12 h	Scaffold2.g181	*noxA*	Superoxide-generating NADPH oxidase heavy chain subunit A	9.14901
	Scaffold26.g60	*pha2*	Prephenate dehydratase	8.255327
	Scaffold3.g212	*QPCT*	Glutaminyl-peptide cyclotransferase	7.622853
	Scaffold12.g184	*SPAC513.07*	NAD-dependent epimerase/dehydratase	7.52363
	Novel.3896	*ALDH6A1*	Methylmalonate-semialdehyde dehydrogenase, mitochondrial precursor	6.657672
	Novel.5121	*bgn13.1*	Endo-1,3(4)-beta-glucanase	5.17254
	Scaffold9.g19	*AIFM2*	Apoptosis-inducing factor	4.56581
	Scaffold4.g409	*FUB8*	Noncanonical nonribosomal peptide synthetase	4.510308
	Scaffold1.g129	*abf1*	Alpha-L-arabinofuranosidase B	4.219728
	Scaffold24.g41	*glcA*	Glucan endo-1,3-beta-glucosidase	4.149159
24 h	Scaffold23.g56		WSC domain-containing protein	6.121065
	Scaffold41.g30	*neg-1*	Endo-1,6-beta-D-glucanase	4.952524
	Novel.5121	*bgn13.1*	Endo-1,3(4)-beta-glucanase	4.795401
	Novel.5464	*ARB_02077*	Glucan endo-1,3-beta-glucosidase	3.911272
	Scaffold11.g98	*cel3A*	Beta-glucosidase	3.879044
	Scaffold12.g101	*eng2*	Endo-1,3(4)-beta-glucanase	3.805877
	Scaffold12.g245	*prb1*	Subtilase	3.391825
	Scaffold18.g111	*mfnA*	L-aspartate decarboxylase	3.217549
	Scaffold7.g216	*YEL023C*	Uncharacterized protein	3.112885
	Scaffold45.g14	*chi2*	Endochitinase	2.711266
48 h	Scaffold23.g56		WSC domain-containing protein	5.267517
	Novel.5121	*bgn13.1*	Endo-1,3(4)-beta-glucanase	4.572012
	Scaffold12.g245	*prb1*	Subtilase	3.663304
	Scaffold41.g30	*neg-1*	Endo-1,6-beta-D-glucanase	3.622379
	Scaffold31.g82	*CUTA*	Cutinase	3.521937
	Scaffold3.g450	*arsH*	NADPH-dependent FMN reductase	3.228058
	Scaffold11.g98	*cel3A*	Beta-glucosidase	3.177845
	Scaffold8.g89	*gatA*	4-aminobutyrate aminotransferase	2.890697
	Scaffold10.g25	*YLL056C*	Uncharacterized protein	2.857343
	Scaffold9.g58	*apdF*	Aspyridones efflux protein	2.820267
72 h	Scaffold4.g229	*rpl-3*	60S ribosomal protein L3	12.9414
	Novel.8070	*FRE7*	Ferric/cupric reductase transmembrane component 7	8.43363
	Scaffold23.g56		WSC domain-containing protein	4.991655
	Scaffold65.g8	*hpcH*	4-hydroxy-2-oxo-heptane-1,7-dioate aldolase	4.481846
	Novel.5121	*bgn13.1*	Endo-1,3(4)-beta-glucanase	4.327306
	Scaffold8.g89	*gatA*	4-aminobutyrate aminotransferase	4.17367
	Scaffold5.g281	*gpd1*	Glyceraldehyde-3-phosphate dehydrogenase	4.130136
	Scaffold9.g58	*apdF*	Aspyridones efflux protein	4.031324
	Novel.7601	*SPAC5H10.01*	Hydro-lyase C5H10.01	3.735899
	Scaffold6.g107	*liuE*	3-hydroxy-3-isohexenylglutaryl-CoA/hydroxy-methylglutaryl-CoA lyase	3.61478

## Data Availability

Not applicable.

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
