# Peer review of "Identification of Mycoparasitism-Related Genes against the Phytopathogen Botrytis cinerea via Transcriptome Analysis of Trichoderma harzianum T4"

_jof, 2023, doi:10.3390/jof9030324_

Round 1

Author Response

Response to Reviewer 1 Comments

Thank you very much for the referee careful review and constructive suggestions with regard to our manuscript. Meanwhile, I am also very grateful to his/her for the specific modification strategy. Those comments are valuable and very helpful to us. We have read through comments carefully and have revised our manuscript to the best of our ability in accordance with the suggestions of the referee. Below is our point-by-point responses to the referee’s comments and suggestions.

Point 1: The referee put forward some suggestions on English expressions, language style and spelling checking and so on.

Response 1: We thank the reviewer for his/her careful reviewing. We agree with all the reviewer's suggestions and have carefully revised our manuscript in revision mode according to the suggestions of the referees.

Point 2:  The referee said “Which plants? Please clarify.

Response 2: Done as requested. We’ve changed [plants] to [tomatoes](page 1, line 12)

Point 3: The referee mentioned that “How you prepare the inactivated cell wall? Did you mean inactivated cells?

Response 3: Thank you for raising this issue. We ignored the problem. Yes, I mean inactivated cells. We have added the preparation conditions and methods of inactivated cell wall of B. cinerea (previously autoclaved at 120 °C for 20 min) in the revised manuscript (Part 2.2, page 3).

Point 4: The referee said “How the non-inoculated medium was used as a control? Did you mean inoculated only with T. harzianum?

Response 4: Thanks for pointing out this problem that the non-inoculated medium was not used as a control, I mean the inoculated only with T. harzianum. We have revised the sentence (Part 2.2, page 3: “the CDB medium containing only sucrosse was used as the control") to “the CDB induction medium inoculated only with T. harzianum grown mycelium without BCCW was used as the control.”(Part 2.2, page 3).

Point 5: The referee suggested “Part 2.3, page 3: Please rewrite this part in more details so other researchers can repeat these steps”.

Response 5: We thank the referee for his/her suggestion. We added as more details as we can in Part 2.3 in the revised manuscript, as suggested by the reviewer, including the cDNA Synthesis, the instrument model, city and country, the software city and country; and the source of the reference genome of T. harzianum T4 (Part 2.3, page 3).

Point 6: The referee said that “18s rRNA, and using α-tubulin which one was used as a reference gene?

Response 6: We thank the reviewer for pointing this out. ​Both 18s rRNA and α-tubulin are internal reference genes. The sentence "18s rRNA, and using alpha-tubulin transcripts as the internal reference gene" is a typo. We apologize for the error. We have modified the typo to “the 18s rRNA and α-tubulin were used as internal references to normalize the amount of total RNA present in each reaction"(Part 2.5, page 4).

Point 7: The referee commented that “Why you used two discs of Trichoderma in the control plate??? The fungus does not antagonize itself”.

Response 7: We thank the reviewer for his/her comment and suggestion. We completely agree with the reviewer that the fungus does not antagonize itself. The expression level of transcripts of Trichoderma varies at different growth stages. In order to reduce the experimental error between the control group and the treatment group and eliminate the differential genes expressed by Trichoderma at different growth stages, we used two discs of Trichoderma in the control plate to ensure that samples were taken at the same stages as the control group (before, during and after physical contact with hyphae). I am very sorry for my imprecise English expressions. We have changed "the T. harzianum T4 antagonized itself as a double-culture control” to “as a control, dual-culture confrontation assays were conducted following the same procedure described above except that T. harzianum was challenged with itself.”(Part 2.6, page 4).

Point 8: The referee mentioned that “Why you used sucrose as carbon source? What is the effect of sucrose on the mycoparasistism?

Response 8: Thanks for the comment. We used sucrose as a carbon source because the carbon is essential for fungi, and sucrose can be used as a carbon source to support the growth and development of T. harzianum T4. Trichderma can use the lysed host cell wall as a source of carbon and nitrogen, so the Czapek-Dox broth (CDB) induction medium with known carbon and nitrogen sources to eliminate the effect of medium with unknown carbon and nitrogen sources on the expression of genes associated with mycoparasitism. Based on the comprehensive comparison of the references, we selected the CDB medium as the induction medium, with sucrose as the carbon source. “What is the effect of sucrose on mycoparasistism?” This is a very good question. We believe that the selection of different carbon sources may affect the expression of genes associated with mycoparasitism. In the next study, we are studying the effect of sucrose or glucose as a carbon source on the growth and development of T. harzianum T4 and on the mycoparasistism.

Point 9: The referee advised “Font of the information on x and y axes needs to be magnified, they are unreadable” in Figure 2.

Response 9: Thanks for raising this important issue. In order to show our results more clearly, we have modified Figure 2 and enlarged the font to ensure that the information can be read. (Part 3.2, page 7, Figure 2)

Point 10: This referee suggested that in Figure 3“The font is unreadable, please modify the figure

Response 10: We thank the referee for his/her suggestion. Sorry for the unreadable font in this figure, we have modified Figure 3 and enlarged the font. (Part 3.2, page 8, Figure 3)

Point 11: The referee advised that about Figure 4 “Modify the font size

Response 11: Thanks for pointing out this problem, the font size of Figure 4 has been modified. (Part 3.3, page 11, Figure 4)

Point 12: The referee commented that about Figure 8 “Where is the statistical analysis for these data? How can i see if the difference in expression is significant or not?” and “Add a clarification for the abbreviations (BC, C, AC) in the figure caption. In addition, you should clarify the full names of the genes”.

Response 12: Thank you for raising this insight question. We have added the statistical analysis and significance analysis in the revised manuscript (Part 2.6, page 5). In addition, in the revised manuscript, we have modified Figure 8(Part 3.7, page 17, Figure 8),  added explanations of abbreviations (BC, C, AC) in the caption of the figure, and clarified the full names of genes in the figure (Part 3.7, page 17).

Reviewer 2 Report

This is a mechanistic study on biocontrol of pathogenic fungi by Trichoderma harzianum. It provides further insights into mycoparasitism, the unique biocontrol action of Trichoderma sp. Differentially expressed genes (DEGs) were identified with RNA-seq analysis of T. harzianum T4 strain after different periods in culture containing inactivated B. cinerea cell wall (BCCW). RT-PCR generally confirmed the RNA-seq results. 24 upregulated genes associated with mycoparasitism were further validated in T. hazianum samples collected during dual culture of T. harzianum T4 and B. cinera. The same genes were claimed to be upregulated (see comments below), hence further supports the involvement of these genes during mycoparasitism.

Overall, this is an interesting mechanistic study of mycoparasitism of Trichoderma and provides good insights into how Trichoderma functions. It has good general interest.

However, the paper is not ready to publish in its current form. Several issues need to be addressed to make it more robust and well-supported:

  1. Inconsistent results of double culture validation: It is a very good idea to validate the DEGs with double culture. The expectations are: a) The selected DEGs should remain upregulated during and after contact; 2) DEGs upregulation is expected higher for during (C) and after contact (AC) samples in comparison with before contact samples (BC). There are some noticeable inconsistencies between the statement (Part 3.7, p17: “based on the RT-qPCR results, all 24 selected genes were upregulated during the dual culture…” and the results in Figure 8, which shows the reduced expression for many genes with a relative expression <1 in one or both C and AC samples. In addition, only 9 genes had increased transcription during contact and after contact over before contact. This is a critical flaw of this paper and needs to be addressed satisfactorily.
  2. Missing some important interesting genes for validation experiment. Based on table 3, the top 10 updated genes vary for the different time periods. The only consistent presence in 3 or 4 time periods are bgn13.1 and neg-1, both encoding endo-beta-glucanases, the enzyme presumably responsible to digest the hyphae of the target fungus. Surprisingly, these two genes were not included in the validation experiment. There is a need for a good explanation.
  3. Variation of DEGs for different time points. Figure 2 gives GO functional categories of the DEGs. A pattern is observed: highest number of DEGs for 24 hr, much lower numbers for 24 and 48 hr and the number came back high for 48 hr. This pattern remains the same for almost all categories. In addition, GO enrichment analysis bubble diagrams (Figure 3) show variable results among the four time periods. The reviewer has some concerns about technical errors. The authors need to elaborate on this discrepancy and confirm that cautions were exercised to eliminate/minimize possible human errors.  

Author Response

Response to Reviewer 2 Comments

The referee commented that: “Overall, this is an interesting mechanistic study of mycoparasitism of Trichoderma and provides good insights into how Trichoderma functions. It has good general interest.” We sincerely thank the referee for his/her support and high appreciation of our work. Below is our point-by-point responses to the referee’s comments and suggestions.

Point 1: Inconsistent results of double culture validation: It is a very good idea to validate the DEGs with double culture. The expectations are: a) The selected DEGs should remain upregulated during and after contact; 2) DEGs upregulation is expected higher for during (C) and after contact (AC) samples in comparison with before contact samples (BC). There are some noticeable inconsistencies between the statement (Part 3.7, p17: “based on the RT-qPCR results, all 24 selected genes were upregulated during the dual culture…” and the results in Figure 8, which shows the reduced expression for many genes with a relative expression <1 in one or both C and AC samples. In addition, only 9 genes had increased transcription during contact and after contact over before contact. This is a critical flaw of this paper and needs to be addressed satisfactorily.

Response 1: Thank you for raising this important issue. First of all, we agree completely with the reviewer“the selected DEGs should remain upregulated during and after contact”. We revised this part in the manuscript includes added a significance analysis of the data (Part 2.6, Page 4), modified Figure 8 (Part 3.7, Page 16), added explanations of abbreviations (BC, C, AC) in the caption of the figure, and clarified the full names of genes in the figure (Part 3.7, page 16). Secondly, the referee commented that “DEGs upregulation is expected higher for during (C) and after contact (AC) samples in comparison with before contact samples (BC)” and “only 9 genes had increased transcription during contact and after contact over before contact”. We apologize for not expressing ourselves more clearly. In dual culture RT-qPCR validation, the before contact (BC) samples is not the control group of the during (C) and after contact (AC) samples. Whether a gene is significantly upregulated in the during (C) and after contact (AC) is not referred to the before contact samples (BC). Due to the existence of volatiles from B. cinerea, some genes were significantly upregulated before contact (BC). T. harzianum was used as the control groups before contact, during and after contact with itself. We have revised the corresponding part of the manuscript (Part 2.6, page 5). As a result, 15 of the 24 selected genes were significantly upregulated during (C) and after contact (AC) between T. harzianum T4 and B. cinerea. Finally, we changed the sentence (Part 3.7, page 15: “based on the RT-qPCR results, all 24 selected genes were upregulated during the dual culture…”) to: “based on the results of dual-culture RT-qPCR, 15 of the 24 genes were upregulated during and after contact between T. harzianum T4 and B. cinerea, providing further evidence that these genes are important factors in mycoparasitism.” (Part 3.7, page 15).

Point 2:Missing some important interesting genes for validation experiment. Based on table 3, the top 10 updated genes vary for the different time periods. The only consistent presence in 3 or 4 time periods are bgn13.1 and neg-1, both encoding endo-beta-glucanases, the enzyme presumably responsible to digest the hyphae of the target fungus. Surprisingly, these two genes were not included in the validation experiment. There is a need for a good explanation.

Response 2: We thank the reviewer for his/her comment and suggestion. We agree with the reviewer that bgn13.1 and neg-1 are considered important, and that endo-beta-glucosidase is presumably responsible for digesting the hyphae of the target fungus. We verified bgn13.1 and neg-1 genes in the double culture verification, but because the expression levels of these two genes were low (the maximum FPKM was less than 15), they were was difficult to verify them successfully by RT-qPCR, so they were not shown in the manuscript. We will further validate the bgn13.1 and neg-1 genes by gene knockout and over-expression in the following studies, as stated in the discussion section of the manuscript.

Point 3:Variation of DEGs for different time points. Figure 2 gives GO functional categories of the DEGs. A pattern is observed: highest number of DEGs for 24 hr, much lower numbers for 24 and 48 hr and the number came back high for 48 hr. This pattern remains the same for almost all categories. In addition, GO enrichment analysis bubble diagrams (Figure 3) show variable results among the four time periods. The reviewer has some concerns about technical errors. The authors need to elaborate on this discrepancy and confirm that cautions were exercised to eliminate/minimize possible human errors.

Response 3: Thanks to the reviewer for helpful comments. After careful verification, we confirmed that there are no technical errors in Figure 2 and Figure 3. However, in order to clearly show the functional category and gene number of the DEGs (including upregulated and downregulated) in T. harzianum T4 grown for 12, 24, 48, and 72 h in the presence of BCCW, we modified Figure 2 (Part 3.2, page 6). GO functional categories of all DGEs (Figure 2) connects DEGs with its corresponding GO function to get the GO annotation information of genes. There are indeed some similar patterns in the GO function categories of DEGs in the four periods. In order to identify which genes were up-regulated to increase the mycoparasitism of T. harzianum T4, we further analyzed the GO enrichment analysis of up-regulated DEGs, and showed the top 20 GO terms with significant enrichment (Figure 3). GO enrichment analysis refers to the enrichment of gene sets with similar GO function functions together by statistical test algorithm, which is convenient for the study of genes with a certain type of GO function. Because the upregulated of DEGs varies in T. harzianum T4 grown for 12, 24, 48, and 72 h in the presence of BCCW, the results in Figure 3 shown variable results among the four time periods. To show the results more clearly, we modified Figure 3 and enlarged the font (Part 3.2, page 7).

Round 2

Reviewer 1 Report

I think the manuscript is acceptable in this form

Reviewer 2 Report

The issues raised have been reasonably/satisfactorily addressed.